# Targeting *N*-Methyl-d-Aspartate Receptors in Neurodegenerative Diseases

**DOI:** 10.3390/ijms25073733

**Published:** 2024-03-27

**Authors:** Allison Carles, Aline Freyssin, Florent Perin-Dureau, Gilles Rubinstenn, Tangui Maurice

**Affiliations:** 1MMDN, University of Montpellier, EPHE, INSERM, Montpellier, France; allison.carles@umontpellier.fr (A.C.); a.freyssin@rest-therapeutics.com (A.F.); 2ReST Therapeutics, 34095 Montpellier, France; florent.perin-dureau@rest-therapeutics.com (F.P.-D.); gilles.rubinstenn@rest-therapeutics.com (G.R.)

**Keywords:** *N*-methyl-d-aspartate receptor, neurodegenerative diseases, Alzheimer’s disease, fluoroethylnormemantine (FENM)

## Abstract

*N*-methyl-d-aspartate receptors (NMDARs) are the main class of ionotropic receptors for the excitatory neurotransmitter glutamate. They play a crucial role in the permeability of Ca^2+^ ions and excitatory neurotransmission in the brain. Being heteromeric receptors, they are composed of several subunits, including two obligatory GluN1 subunits (eight splice variants) and regulatory GluN2 (GluN2A~D) or GluN3 (GluN3A~B) subunits. Widely distributed in the brain, they regulate other neurotransmission systems and are therefore involved in essential functions such as synaptic transmission, learning and memory, plasticity, and excitotoxicity. The present review will detail the structure, composition, and localization of NMDARs, their role and regulation at the glutamatergic synapse, and their impact on cognitive processes and in neurodegenerative diseases (Alzheimer’s, Huntington’s, and Parkinson’s disease). The pharmacology of different NMDAR antagonists and their therapeutic potentialities will be presented. In particular, a focus will be given on fluoroethylnormemantine (FENM), an investigational drug with very promising development as a neuroprotective agent in Alzheimer’s disease, in complement to its reported efficacy as a tomography radiotracer for NMDARs and an anxiolytic drug in post-traumatic stress disorder.

## 1. Introduction

Neurotransmission within the central nervous system (CNS) is essentially based on two main neuronal systems that coordinate the excitatory and inhibitory inputs to establish and maintain a constant excitation/inhibition ratio. Glutamate (Glu) is the most important excitatory neurotransmitter in the CNS, with 50–70% utilization by synapses [1,2]. Glu binds to two types of receptors, namely metabotropic Glu receptors (mGluRs), which belong to the G protein-coupled receptor (RCPG) superfamily, and ionotropic Glu receptors, including α-amino-3-hydroxy-5-methyl-4-isoxazolepropionic acid receptor (AMPAR), *N*-methyl-d-aspartate receptor (NMDAR), and kainate receptor. These receptors are present in the whole CNS, with the highest densities in cortical and hippocampal structures [3,4]. They are involved in neurodevelopmental processes, cognitive functions, learning, memory, and synaptic plasticity [5,6,7]. NMDARs are localized in the synaptic and/or extra-synaptic membranes [8], depending on the neuronal maturation stage and receptor function, with complex inter-relations between the two sets of receptors. Extra-synaptic NMDARs represent 75% of NMDARs in immature hippocampal neurons and are still expressed during adulthood, while synaptic NMDAR expression gradually increases throughout brain development [9,10,11,12].

Glu binding to NMDARs leads to an increase in intracellular calcium (Ca^2+^) which acts as a second messenger within the synapse by inducing signaling pathways necessary for neurotransmission and brain plasticity [13,14]. Current flow through the NMDAR channel is largely blocked by external Mg^2+^ ions at resting membrane potential, but it can be relieved by depolarization [15]. The receptor activity is modulated by different molecules acting as co-agonists, like Gly and D-Ser, which bind to a specific site on NMDARs [16]. It is now accepted that the induction of long-term potentiation (LTP) and long-term depression (LTD), in the CAl layer of the hippocampus, is highly dependent on NMDAR activation at the post-synaptic synapse [17,18]. Astrocytes reuptake about 90% of the Glu released in the synaptic cleft, through sodium-dependent Glu transporters, named excitatory amino acid transporters (EAATs), and approximately 20% of the synaptic Glu released effectively reaches the post-synaptic Glu receptors, while the rest interacts with extra-synaptic receptors [19]. When uptaken by glial transporters, Glu is metabolized and recycled by glutamine synthetase into glutamine and redistributed to the pre-synapse for recycling. Moreover, high-resolution microscopy techniques revealed that NMDARs are not randomly distributed at synapses, but organized into finely regulated nanoscale signaling domains that can be remodeled by rapid diffusion movements within the plasma membrane. This property sustains the ability of synapses to quickly adapt and change their composition, therefore facilitating information encoding [20].

NMDARs are currently the targets of numerous drug discovery programs. Indeed, there are still numerous uncovered medication opportunities for the treatments of neurologic and neuropsychiatric disorders, and drug development in brain disorder indications is highly challenging. The limitations of pertinent animal models for preclinical studies and methodological limitations to understand the functioning of NMDAR in vivo in physiological or pathological conditions impede rapid progress. NMDARs’ pathological activation is a main mediator of neuronal injuries or dysfunctions in neurodegenerative diseases such as Alzheimer’s disease (AD), Huntington’s disease (HD), and Parkinson’s disease (PD), and, outside the focus of this review, in neuropsychiatric disorders such as post-traumatic stress disorder (PTSD), epilepsy, and schizophrenia [21,22,23,24]. In neurodegenerative conditions, an excess of Glu release resulting from synaptic alterations massively increases the intracellular Ca^2+^ influx. The normal blockade by Mg^2+^ of the ionophore is removed, and NMDAR activity is pathologically enhanced. Since the calcium permeability is relatively high, extra-synaptic NMDA receptors are preferentially targeted in these conditions and drive neuronal excitotoxicity [25].

In the present review, we will provide a concise assessment of the role and impact of NMDA systems in neurodegenerative diseases, particularly in AD, HD, and PD, in terms of NMDAR structure, distribution, localization, and regulation at the glutamatergic synapse in physiological conditions and neurodegenerative diseases. We will discuss the mechanism of action of NMDAR-targeting drugs and highlight a novel uncompetitive antagonist, fluoroethylnormemantine (FENM), that showed several benefits over memantine in AD models and over ketamine in PTSD models.

## 2. Physiology of NMDA Receptors

### 2.1. Structure, Composition, and Localization

The NMDAR is a transmembrane protein supercomplex with four domains: the extracellular amino-terminal domain (ATD), the extracellular ligand-binding domain (LBD), the transmembrane domain (TMD), and the intracellular carboxyl C-terminal domain (CTD) [26]. Structurally, functional NMDARs are heterotetramers (≈1.5 KDa) composed of two GluN1 subunits bearing the Gly or D-Ser binding sites, two GluN2A~D subunits bearing the Glu binding site, and a GluN3 subunit (Figure 1) [27,28]. Subunits differ by the length of the GluN CTD region [29]. Non-competitive binding sites are localized within the ion channel pore. First, Mg^2+^ exerts a voltage-dependent blockade of the ion flux, dependent on an asparagine residue in the second transmembrane segment. Depolarization results in Mg^2+^ ion removal and the influx of Ca^2+^. Second, phencycline (PCP) and PCP-like drugs bind within the pore and exert an uncompetitive antagonism of NMDAR and are often referred to as channel blockers. The PCP site is accessible when the receptor is in an activated state [30,31,32]. Other reference uncompetitive antagonists include PCP derivatives such as thienylcyclidine (TCP) and ketamine and chemically unrelated molecules such as dizocilpine and memantine. Third, Zn^2+^ inhibits NMDARs’ current by a high affinity binding in the N-terminal domain of GluN2A [33], inducing a reduction in the channel’s opening probability. The NMDAR subunits are encoded by seven genes. The GluN1 subunit is encoded by a unique gene, GRIN1, but has eight distinct isoforms (a–h, with different splice variants of a single gene). The GluN1 associations with GluN2A~D subunits, encoded by two out of four GRIN2 genes, display specific spatio-temporal expressions in the brain [26]. In addition, GluN3A~B is encoded by two GRIN3 genes. Compelling evidence indicated that diheteromers and triheteromers coexist within a single cell or even at a single synapse, adding to the functional diversity of the post-synaptic responses [26]. In the adult hippocampus and cortex structures, there is a predominant expression of diheteromeric GluN1/GluN2A or GluN1/GluN2B receptors. Moreover, triheteromeric GluN1/GluN2A/GluN2B receptors represent, in these structures, between 15% and 50% of the total receptor population [34,35,36,37]. Although there is 70% identity in GluN2A and GluN2B sequences, the subunits play different roles in the physiological or pathological processes [38]. The distribution is also highly specific, with a predominance of GluN2A subunits at the post-synapse vs. GluN2B subunits at the extra-synaptic membrane [28]. However, NMDARs are mobile in the plasma membrane and GluN2B-containing receptors can move through lateral diffusion between synaptic and extra-synaptic spaces.

GluN3A expression predominates during postnatal development in many brain regions, including the CA1 hippocampus. GluN3A expression declines in the second and third postnatal week and remains low into adulthood throughout most of the central nervous system. GluN3A and GluN3B show a more heterogeneous expression, inducing smaller NMDA currents with reduced calcium permeability in the post-synapse. These subunits therefore act in a dominant-negative manner to block receptor activity. GluN3A is localized in dendritic spines of glutamatergic neurons and influences synaptic stability, receptor assembly, and the functional activity in the neuron [39,40]. Moreover, GluN1/GluN3A NMDARs are functionally expressed in native neurons during development and mediate the excitatory effect of Gly [41]. NMDARs with GluN2C/2D subunits display a lower sensitivity to Mg^2+^ ions and slower deactivation kinetics as compared with GluN2A/2B-containing NMDARs. The expression of GluN2C/2D has been reported in the adult forebrain with, particularly, GluN2D subunits playing a major role in the synaptic transmission of hippocampal neurons [42,43].

The cellular distribution of NMDARs is also of importance and, in the physiological concept of a “tripartite synapse” (Figure 2), which is based on direct communication between the neuron, the pre-synaptic terminal, the post-synaptic spine, and peri-synaptic astrocytes, NMDARs are expressed in all cell types with specific roles and impacts on synapse efficiency [44]. NMDARs were also described in oligodendrocytes, which play a major regulatory role on glucose transport and axonal metabolism [45,46]. The existence of functional glial NMDARs shows specificities, notably with an activation at negative membrane potential, and differ from neuronal NMDARs not only by a weaker Mg^2+^ block but also by lower calcium permeability. Transcripts for all seven NMDAR subunits have been found in cultured human and rat astrocytes. The permeability of calcium in cortical mouse astrocytes is about three to four times lower than in neurons [47]. Astrocytic NMDARs also play an active role in the glutamatergic synapse [48], and astrocytic activation increases calcium release paired with post-synaptic depolarization-induced LTP [49,50,51]. Moreover, microglia express functional NMDARs and, in microglial culture models, treatment with NMDAR antagonists prevented NMDA-induced microglial proliferation and reduced morphological activation and the release of pro-inflammatory cytokines [52].

### 2.2. NMDARs in the Glutamatergic Synapse

Glu, synthetized from glutamine and sequestered within synaptic vesicles, is released from the pre-synapse into the synaptic cleft and binds to the NMDAR in the post-synapse (Figure 1). Only 10% of Glu present in the synaptic cleft (~1.1 mM) reach Glu receptors within approximately 1.2 msec. The other 90% is reuptaken by the EAAT-1 transporters in astrocytes and metabolized into glutamine by glutamine synthase. Glutamine can be released into the synaptic space by the sodium-coupled neutral amino acid transporter (SNAT3/5). SNAT1/2 transporters on pre-synaptic neurons uptake glutamine and predominantly regulate neuronal glutamine uptake [53,54]. In response to the synaptic release of Glu, NMDARs are activated within hundreds of msec, and activation is maintained for a long time after all Glu has been removed from the synaptic cleft. In response to Glu release, AMPARs induce a depolarization in the post-synapse and allow Na^+^, K^+^, and mainly Ca^2+^ to flow through the NMDAR pore [26,55]. At negative membrane potentials, or near the resting membrane potential, Mg^2+^ is present in the NMDAR pore and prevents ion fluxes; but at positive membrane potentials, Mg^2+^ is released through depolarization allowing for an important ion flux with a large outward current. The activation of NMDARs lasts much longer than AMPARs that close rapidly within a few msec. On the one hand, the induction of LTP requires the activation of synaptic NMDARs and large increases in intracellular [Ca^2+^]_i_. On the other hand, the induction of LTD requires the internalization of synaptic NMDARs, the activation of extra-synaptic NMDARs, and lower increases in [Ca^2+^]_i_. Moreover, the induction of LTP leads to the recruitment of AMPARs and dendrite growth, while the induction of LTD is more related to spine shrinkage and synaptic loss.

When penetrating the post-synaptic cell, Ca^2+^ binds to calmodulin and activates calmodulin kinase II (CaMKII), which leads to kinase autophosphorylation at Thr^286^ [56]. CaMKII is a highly abundant protein in post-synaptic density, which phosphorylates AMPAR in the GluA1 subunit. This effect increases AMPAR conductance and binding to post-synaptic density protein 95 (PSD-95), a pivotal scaffolding protein at excitatory synapses, and leads to the translocation of the internalized AMPAR to the post-synapse [57,58]. PSD-95 also regulates NMDAR activity [59,60,61]. At the same time, the increase in intracellular Ca^2+^ activates other kinase pathways such as Ras-Raf, mitogen-activated protein kinases (MAPK family: Map3K, Map2K), and extracellular signal-regulated protein kinase 1/2 (ERK1/2). The phosphorylation of ERK1/2 in the cell nucleus generates the phosphorylation of cAMP response element-binding protein (CREB) at Ser^133^, and in turn, gene transcription [62,63]. Intracellular Ca^2+^ also activates adenylate cyclase or guanylate cyclase, leading to increases in 3′,5′-adenosine monophosphate and/or 3′,5′-guanosine monophosphate (cAMP and/or cGMP). The induction of cAMP and/or cGMP activates protein kinase A (PKA), PKG, or PKB (AKT). Activations of PKA or AKT, by phosphorylation on Thr^197^ or Thr^308^, respectively, also contribute to CREB phosphorylation on Ser^133^. CREB phosphorylation consequently promotes cellular plasticity by regulating the expression of mRNA levels of CREB target genes such as c-fos or Nr4a1 different proteins, like tissue-type plasminogen activator (tPA) or brain-derived neurotrophic factor (BDNF). These proteins indeed play a major role in LTP, memory processes, and anxiety-related behaviors. tPA, a serine protein, facilitates the conversion of pro-BDNF to mature BDNF, a major regulator of brain plasticity [64,65]. These cellular pathways are predominantly involved in antioxidant defense, neuroprotection, memory processes, and hippocampal LTP.

**Figure 2 ijms-25-03733-f002:**
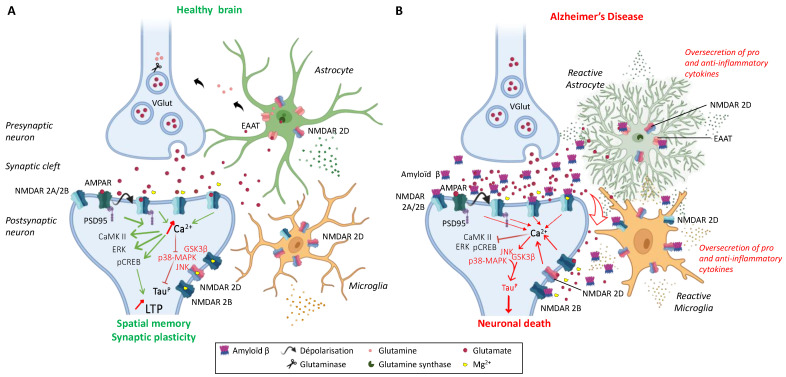
The tripartite synapse comprising pre- and post-synaptic glutamatergic neurons, astrocytes, and microglia in (**A**) a healthy brain and (**B**) an AD patient brain. Glutamatergic excitotoxicity is induced by the accumulation of amyloid-ß aggregates in the synaptic cleft, interacting with post-synaptic NMDARs and astrocytic EAATs, leading to synaptic loss and impaired physiological and behavioral responses. Adapted from [26,43,62].

Signal diminution or the weakening of network connection is also important in the brain. To ensure circuit remodeling, encoding, or even the erasure of memory, LTD processes contribute to maintain synaptic efficiency. In the case of weak Glu release from the pre-synapse, or modest depolarization in the AMPAR and NMDAR at the post-synapse, a weak Ca^2+^ influx through the NMDAR leads to the weak activation of PP2B or PP1 phosphatases and mitochondrial caspase-3. Ultimately, it could lead to apoptosis by a mechanism that involves in the dephosphorylation of AMPAR. This phenomenon leads to the endocytosis of AMPAR, a degradation by lysosome, and LTD [55,66,67]. Electrophysiologically, LTD could be induced using a repetition of the activation of the pre-synaptic synapse at low frequency without post-synapse activation. In physiological conditions, with neurons at their resting membrane potential and sub-maximal Mg^2+^ block of NMDAR, a Ca^2+^ influx could be induced in response to low-frequency synaptic stimulation [55,68]. In rodent models, a stimulation at 1 Hz for 15 min is typically used to induce LTD acutely in hippocampal slices [69].

### 2.3. Regulation of NMDAR Activity

Data suggested that NMDARs can be internalized quickly after physiological triggers [70,71] or that a switch in the NMDAR subunit composition can happen during early neuronal development [27,37,72]. For instance, in the cortex, GluN2B is more expressed than GluN2A in the early postnatal brain, and there is a shift during development with a progressive increase in the GluN2A/GluN2B ratio. This switch could be driven by sensory experience during brain development [73] and is related to an increase in GluN2A mRNA levels, which are driven by NMDAR activity.

In basal conditions, AMPARs (GluA1/2) and NMDARs (GluN2A/2B) are expressed at the post-synapse with the PSD-95 scaffold protein. An important regulation is achieved by NMDAR and AMPAR trafficking in synapses during LTP [58]. In the presence of intra-cellular Ca^2+^, AMPARs are immobilized in the post-synapse, in a CaMKII-dependent manner, and anchored by PSD-95. For NMDAR, GluN2A is tightly anchored at the synapse as it allows for the entry of Ca^2+^ in the developing hippocampus [74], allowing for the activation of post-synaptic kinase cascades, while GluN2B is more diffusible in immature neurons, allowing for the diffusion and/or redistribution of more intracellular actors like CaMKII or casein kinase 2 [75,76,77]. The Ca^2+^ increase allows for the induction of LTP and the concentration of AMPARs in post-synapse. This phenomenon is very important during the initial phase of synaptic potentiation and for the recycling by exocytosis of receptors to the synapse. Moreover, AMPARs diffuse laterally in the cell surface when it is not bound to PSD-95. The phosphorylation of Thr^321^ in the PDZ domain in the γ2 TARP auxiliary subunit of PSD-95 can separate in C-terminal PSD-95 PDZ domains and allows for the diffusion of AMPARs on the cell membrane regulating LTP induction [78,79,80,81]. Some studies highlighted that AMPAR trafficking did not impact NMDAR-mediated LTP, but the reverse is true, suggesting that NMDAR expression at the post-synapse could be due to the exocytosis of NMDAR GluN2A, induced by the phosphorylation of SAP97 by CaMKII at Ser^39^ in the endoplasmic reticulum [58,82,83,84]. Thus, this complex drives GluN2A to the post-synapse compartment in the developing hippocampus and allows for GluN2A expression at the post-synapse to mediate plasticity.

The regulation of NMDARs also relies on the phosphorylation of its subunits. For example, it was shown that the GluN1 subunit is phosphorylated by PKC on two residues, Ser^890^ and Ser^896^. The Ser^890^ phosphorylation disrupted the clustering of the GluN1 subunit, while the Ser^896^ phosphorylation did not affect clustering. Interestingly, the phosphorylation of Ser^896^ by PKA appeared to be necessary to increase NMDAR surface expression at the post-synapse [85,86]. The regulation of the intracellular trafficking of GluN1 is also directed by Ser^896^ and Ser^897^ hyperphosphorylation in the endoplasmic reticulum and Golgi apparatus [87]. Among other NMDAR subunits, GluN2A can be phosphorylated on three tyrosines, Tyr^1325^, Tyr^1292^, and Tyr^1387^, resulting in a potentiation of NMDAR currents in the synapse [88]. The phosphorylation of Ser^1048^ of GluN2A regulated the surface expression of GluN1/GluN2A and NMDAR currents. It was shown that the dual-specificity tyrosine-phosphorylation-regulated kinase 1 (DYRK1A)-dependent phosphorylation of GluN2A at Ser^1048^ blocked GuN1/GluN2A internalization that allowed for the expression of more GluN1/GluN2A receptors at the surface. This phosphorylation resulted in the potentiation of NMDAR currents [89]. For GluN2B, the phosphorylation of Ser^1303^ affected synapse distribution and activation [90]. GluN2B also bears three Tyr as phosphorylation sites: Tyr^1252^, Tyr^1326^, and Tyr^1472^. These residues are phosphorylated by Fyn, a kinase expressed in the post-synapse. The phosphorylation of Tyr^1472^ is notably controlled by striatal-enriched tyrosine phosphatase, which belongs to a family of protein tyrosine phosphatases in the glutamatergic synapse. The action of these phosphatases is to increase the endocytosis of the NMDAR at the post-synapse. Conversely, the phosphorylation of Tyr^1472^ stabilizes the NMDAR at the post-synapse, while the phosphorylation of Ser^1480^ induces a destabilization of the NMDAR and involved a decrease in NMDAR expression [91,92]. Numerous studies showed that the phosphorylation of the GluN2A or GluN2B subunit reflected NMDAR activity [93,94].

### 2.4. NMDARs Modulators

NMDAR activity can be modulated by small molecules acting through different mechanisms (Table 1). Among competitive NMDAR antagonists, the highly selective prototype drug is d-2-amino-5-phosphonovaleric acid (d-AP5). d-AP5 inhibits the excitatory response and blocks plasticity (LTP) in GluN2A subunit-containing NMDAR in rodents, thus having an impact on learning at the behavioral level [95,96,97].

Phencyclidine, a dissociative anesthetic with strong addictive properties, binds to a specific site within the ionophore of the NMDAR and acts as a channel blocker. PCP induces psychotic and dissociative schizophrenia-like symptoms resulting from the impairment of NMDAR neurotransmission in vivo [98,99]. Its derivative, ketamine, also acts as an uncompetitive NMDAR antagonist and is still used as an anesthetic, antidepressive, and antihyperalgesic. Ketamine applied in the post-synapse induced an inhibition of the excitatory pyramidal neuron in the extra-synaptic GluN2B subunit. When applied in the pre-synapse, the drug induced an inhibition of the GluN2D subunit in interneurons and provoked a disinhibition of Glu release in the post-synapse. At the same time, it induced an up-regulation of hippocampal AMPARs (GluA1/GluA2). This phenomenon has a consequence on plasticity and sustains its rapid antidepressant efficacy [100,101]. Among other uncompetitive antagonists acting at the PCP site, dizocilpine, also known as (+)-MK-801 maleate, is a nanomolar affinity anticonvulsant with strong amnesic and ataxic properties [102].

**Table 1 ijms-25-03733-t001:** Listing of the reference NMDAR modulators.

Compounds	Modulator	Selectivity	Action	References
D-AP5 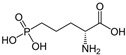	Competitive antagonist	GluN2A	Inhibits excitatory response. Impacts behavioral learning; blocks plasticity (LTP).	[95,96,97]
Riluzole 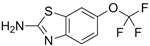	Competitive antagonist	GluN1/GluN2B	Indirect block of NMDAR. Protects from motor deficit.	[103,104,105]
Phencyclidine 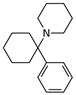	Selective uncompetitive antagonist	GluN2B/D(PCP site)	Induces psychotic and dissociative schizophrenia-like symptoms. Impairs NMDAR neurotransmission in vivo.	[98]
Ketamine 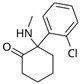	Uncompetitive antagonist	GluN2B/D(PCP site)	Applied in post-synapse: inhibits excitatory pyramidal neuron in extra-synaptic GluN2B. Applied in pre-synapse: inhibits GluN2D in interneuron (induces disinhibition of Glu release in post-synapse). Up-regulates hippocampal AMPARs (GluA1/GluA2). Antidepressant.	[100,106]
Dizocilpine 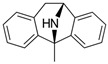	Uncompetitive antagonist	GluN2A/B/D(PCP site)	Anticonvulsant, antidepressant. Induces memory impairments.	[102,107]
Ifenprodil 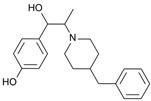	Uncompetitive antagonist	GluN1/GluN2B(N-Terminal domain)	Blocks GluN2B (140-fold preference for NR2B over NR2A subunits). Induces inhibition of GluN2R receptor currents. Anti-Parkinsonian effect.	[108,109,110]
Memantine 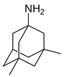	Uncompetitive antagonist	GluN1/GluN2B	Blocks GluN2B extra-synaptic and induces glutamatergic excitotoxicity. Used for moderate-to-severe AD.	[111,112]
Amantadine 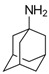	Uncompetitive antagonist	GluN1/GluN2B	Blocks GluN1/GluN2B by accelerating channel closure during channel block. Used as anti-Parkinsonian drug.	[113]
Dextromethorphan 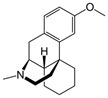	Uncompetitive antagonist	GluN2A	Blocks GluN2A subunit. Prevents neuronal damage and modulates pain sensation	[114,115,116,117,118]

Ifenprodil is an uncompetitive antagonist binding specifically on GluN2B (140x preference for GluN2B vs. GluN2A). It was shown that 150 nM of ifenprodil induced a 75% inhibition of GluN2B receptor currents in human embryonic kidney (HEK-293) cells [119,120]. Another blocker on the GluN2B subunit receptor, memantine, was shown to improve cognitive functions and enhance behavioral disturbance alone as monotherapy or in combination with donepezil for a moderate-to-severe form of AD [121]. Amantadine, a demethylated analogue of memantine, was initially proposed to affect dopaminergic systems before being identified as a blocker of the NMDAR ion channel. The very rapid dissociation kinetics of amantadine in comparison with memantine allowed for it to unblock the ionophore during the brief depolarization associated with the action potential, a characteristic that may help to alleviate deleterious clinical effects [113]. Dextromethorphan is an uncompetitive NMDAR antagonist with similar properties as ketamine and phencyclidine. NMDAR antagonism has a major impact on catecholamine reuptake. Dextromethorphan indeed inhibits the reuptake of serotonin [122]. This property explains dextromethorphan’s high abuse and misuse potential. Although the role of dextromethorphan as an NMDAR antagonist appeared attractive, its clinical use in the treatment of pain in cancer patients led to controversial results [117,118]. Riluzole has neuroprotective, anticonvulsant, anxiolytic, and anesthetic properties. It decreased glutamatergic transmission via NMDAR antagonism and the inhibition of a protein kinase C (PKC) [105]. The drug showed a strong anti-cataleptic potential with beneficial effects in the treatment of muscle rigidity [123].

Finally, antagonists of the GluN2C and GluN2D subunits also exist, making it possible to determine their localization in the brain, particularly in astrocytes but also in neurons [47].

## 3. The Impact of NMDARs in Neurodegenerative Diseases

### 3.1. Alzheimer’s Disease

According to the World Health Organization, there are currently more than 55 million people living with dementia worldwide. Every year, 10 million new cases are detected. This dementia results from different injuries and pathology. More than 60–70% of total cases involve people with AD. It is an aging-related neurodegenerative disorder characterized by the slow deterioration of cognitive functions, such as autonomy, declarative memory, and recognition memory. AD is a progressive disorder, asymptomatic in its early stages, but whose symptoms progress from mild cognitive impairment to severe dementia. Histopathologically, AD is characterized by the extracellular accumulation of aggregated amyloid-β (Aβ) proteins in the hippocampus and cortex and by intraneuronal fibrillar tangles composed of the hyperphosphorylated tau protein. Cerebral neuroinflammation is also massive and characterized by reactive gliosis (microgliosis and astrogliosis), oxidative stress, and the loss of synapses and neurons in layers III/IV of the neocortex, hippocampus, and cortex [124,125,126,127,128]. The increase in oligomeric Aβ may indirectly cause a partial block of NMDARs and induce a shift in the activation of NMDAR-dependent signaling cascades leading to the induction of LTD and synaptic loss [129,130]. The consequence is a progressive cognitive decline, due to deficient cholinergic neurotransmission and excitotoxicity in glutamatergic synapses. Moreover, Aβ oligomers were described to accumulate in the AD patient brain, or in vitro in human cortex neuronal cultures, at GluN2B subunit-containing NMDAR excitatory synapses, inducing a high level of Glu in the synaptic cleft (Figure 2). Moreover, oligomeric Aβ blocks Glu reuptake by the EAATs in astrocytes and activates mGluR in the post-synapse [131], resulting in oxidative stress and apoptosis. The mGluR over-activation leads to AMPAR internalization, the desensitization of NMDARs, and the endocytosis of GluN2B in the post-synapse resulting in impaired LTP and increased LTD and finally, to more neuroinflammation [132,133,134]. In the extra-synapse, the binding of Glu to NMDAR with a GluN2B subunit generates a massive extra-synaptic Ca^2+^ entry and activates deleterious signaling pathways involving MAPK, GSK-3β, or JNK. Activations of these pathways result in an increase in apoptosis and the hyperphosphorylation of the tau protein [26,135]. Moreover, Aβ deposits increase Fyn kinase activity, which phosphorylates GluN2B in Tyr^1472^. This GluN2B phosphorylation increases the subunit association with its scaffold PSD-95 and causes a disruption of synaptic plasticity and LTP in the brain leading to the death of glutamatergic neurons [91,136]. The massive entry of Ca^2+^ also concurs to inhibit signaling pathways beneficial to LTP, involving CAMKII, calcineurin, pCREB, and ERK, to reduce cellular survival factors, including BNDF and CREB, contributing to LTP/LTD alterations [137,138,139]. The activation of extra-synaptic NMDARs leads to an increase in the production of Aβ [140]. In a post-mortem AD human brain, the levels of GluN1, GluN2A, and GluN2B were reported to be decreased in the hippocampus and entorhinal cortex [141,142,143,144,145]. GluN2C and GluN2D mRNA levels did not change in AD patient brains [143].

The prevalence of AD is twice as high in females than in males [146], and besides gender-related sociocultural differences, recent preclinical and clinical data accumulate to show that genetic, epigenetic, and hormonal differences between males and females likely explain gender-related differences in patient susceptibility to develop AD with an earlier onset or with more abrupt disease progression [147]. NMDA systems are not differentially expressed between males and females; however, recent studies suggested that levels in L-Ser are differentially affected in the pathology [148]. L-Ser is the precursor of D-Ser and Gly, the main co-agonists of synaptic and extra-synaptic NMDARs, respectively [149], and mainly synthetized in a sex-dependent manner, by astrocytes [150]. A multi-omics analysis showed that among the pathways differentially affected in female AD patients vs. males, the tricarboxylic acid cycle, ErbB signaling, and HIF-1/insulin pathway are increased while the L-Ser pathway is decreased [150], with putatively direct impacts on NMDAR activity. The NMDA system may therefore be impacted by sex-dependent effects and, in return, could directly participate in the different pathological evolutions observed in male and female AD patients.

Clinical trials showed a significant efficacy of memantine over the placebo in moderate-to-severe forms of AD with an improvement in cognitive functions and activities of daily living [151,152]. Memantine was well tolerated by the patient but showed some adverse side-effects like dizziness, headache, and constipation [153]. The drug was approved by the FDA in 2003.

### 3.2. Huntington’s Disease

In the world, the prevalence of Huntington’s disease (HD) is approximately five cases per 100,000 individuals [154]. HD is an autosomal dominant neurodegenerative disorder characterized by the onset of motor symptoms due to brain atrophy, the degeneration of striatal neurons in the caudate nucleus, putamen, and cerebral cortex with the specific loss of efferent medium spiny neurons [155]. Patients are affected around 35–50 years of age, and symptoms include movement disorders, dementia, cognitive impairment, and behavioral or psychiatric manifestations [156]. HD occurs due to pathological expansions of cytosine-adenine-guanine (CAG) repeats, between 10 and 35, in exon 1 of the huntingtin gene on chromosome 4p16.3. The gene encodes the huntingtin (Htt) protein that therefore shows in HD an elongation of polyQ generating mutant huntingtin (mHtt) [157]. There is a strong correlation between the length of the polyQ repeats and the age of the disease onset [158]. The Htt protein is necessary for the formation of cortical and striatal excitatory synapses, the embryonic shaping of the nervous system, protein trafficking, post-synaptic signaling, vesicle transport, transcription factor regulation, and the regulation of cell death [159,160]. Htt is localized in the cytoplasm and associated with vesicle membranes by the scaffold PSD-95 [161]. mHtt aggregates in the brain, in the nucleus and cytoplasm, and this accumulation results in alterations of gene transcription, neurotransmitter metabolism, and alterations of the expression and distribution of BDNF and its TrkB receptor. Post-mortem studies showed a loss of striatal NMDARs in early symptomatic and pre-symptomatic stages of the disease [162,163,164]. In the R6/2 transgenic mouse model of HD, the expression of GluN2A and GluN2B, but not GluN1, decreased in the striatal region [165,166]. In YAC128 mice, no change in GluN1, GluN2A, or GluN2B was reported in the striatum of mice presenting symptoms, but a decrease in the GluN1 phosphorylation at Ser^897^, previously reported to decrease NMDAR currents, was observed [167].

The pathological process resulted in an imbalance between Glu and dopamine in the striatum, with an increase in dopamine in the post-synapse and an alteration of the Glu reuptake by the glial transporter in corticostriatal synapses. Glutamine mRNA was reduced in rodent models in HD in relation to motor dysfunction [168]. Glutamatergic alterations therefore included the following: (i) an accumulation of Glu in the synaptic cleft, (ii) a decrease in post-synaptic NMDARs, and (iii) an increase in GluN2B extra-synaptic expression in striatal spiny neurons that could be responsible for cell death, when GluN2A is decreased [169,170]. Glutamatergic NMDAR neurotransmission promotes the expression of anti-apoptotic factor Bcl-2, antioxidants, and of the pro-survival trophic factor BDNF. In HD conditions, BDNF release and TrkB activation are decreased in the post-synapse, together with increases in the pro-apoptotic factor Bax, pro-death genes such as FOXO and FBS, the dysregulation of mitochondrial calcium, and CREB-inactivating dephosphorylation signals [171,172]. mHtt indeed contributed to suppress the CREB gene by sequestering its coactivator CBP [173]. Current clinical developments focus on gene therapies and antisense oligos. Alternatively, symptomatic treatments rely on neuroleptics, dopaminergic inhibitors, such as tetrabenazine, and antidepressants adapted to reduce motor and psychiatric signs and sleep disorders [174].

### 3.3. Parkinson’s Disease

Parkinson’s disease (PD) is the second most frequent neurodegenerative disorder and represents a large health burden to society. Approximately 1% of the population over 60 years of age is affected. PD is a progressive neurodegenerative disorder that occurs due to the loss of dopaminergic neurons in the substantia nigra pars compacta (SNc) and resulting in the accumulation of intracytoplasmic inclusions, known as Lewy bodies [175]. A series of alterations in the circuitry of basal ganglia nuclei leads to severe motor control impairments such as tremors, muscular rigidity and bradykinesia [176]. Pharmacological approaches to PD predominantly target the dopaminergic system, notably with dopamine replacement by its precursor, L-3,4-dihydroxyphenylalanine (L-Dopa) [177]. However, prolonged treatment leads to the development of motor complications, known as L-Dopa-induced dyskinesia, involving choreic movements, dystonia, and ballism [178]. NMDARs are largely regulated by dopaminergic afferents and very abundant in the basal ganglia to control the release of the neurotransmitters γ-aminobutyric acid (GABA) and acetylcholine [179]. NMDAR antagonists could be used to decrease L-Dopa dosage and diminish any potential oxidative damage, since oxidative products of dopaminergic neurons play a key role in cell death [180]. In addition, the expression and activity of Glu receptors are also affected during L-Dopa-induced dyskinesia [181]. NMDAR antagonists, such as dextromethorphan, have been reported to suppress dyskinesia in PD patients. Adverse effects at high doses, however, limit this treatment strategy [182]. Alternatively, amantadine remains an interesting therapeutic option in the management of L-Dopa-induced dyskinesias [183]. Amantadine was associated with an increased lifespan in patients with PD, suggesting that it may have neuroprotective properties [184].

Memantine has also been tested clinically in PD patients with moderate success, as it did not appear to share the anti-dyskinetic activity of amantadine [185]. A second generation of adamantane-based drugs is being designed, seeking to improve the clinical efficacy [186].

## 4. Fluoroethylnormemantine (FENM): A New Generation NMDAR Uncompetitive Antagonist

### 4.1. ^18^F-FENM as a PET NMDAR Radiotracer

Memantine and several derivatives have been fluorinated with a positron-emitting isotope ^18^F and tested as radiotracers of positron emission tomography (PET) for the in vivo labeling of NMDARs [187,188,189]. ^18^F-memantine was homogeneously distributed in the cortex and basal ganglia regions, as well as the cerebellum. However, the observed pattern of ^18^F-memantine uptake in the whole brain was not consistent with autoradiographic studies performed on post-mortem human brains with another NMDAR radiotracer, ^3^H-TCP [190], and did not reflect the regional NMDAR distribution [187,188,189]. Moreover, the radioligand binding was partly displaced by haloperidol, suggesting some binding at sigma-1 receptors [189]. Therefore, the radiotracer did not appear suitable for the PET imaging of the NMDA receptors. Among derivatives, ^18^F-fluoroethylnormemantine (^18^F-FENM, Figure 3) showed an excellent selectivity and specificity for NMDAR in preclinical models in vivo. ^18^F-FENM effectively crosses the blood–brain barrier after intravenous administration and to bind to the grey matter, cerebellar cortex, and central grey nuclei [191,192]. Its specific distribution matched the one of the GluN1 subunit, and a low non-specific binding level was seen after pre-injection with ketamine at anesthetic doses. Moreover, FENM competed in vitro with ^3^H-TCP in rat brain membranes with a Ki of 3.5 µM [193]. As observed for memantine, FENM was poorly metabolized in vivo with a good stability in plasma and the level of plasma protein binding. However, as compared with other PET radiotracers, it showed a low effective dosimetry dose [192]. Moreover, the tracer was used recently in a preclinical model of excitotoxicity induced in Sprague Dawley rats by stereotaxic quinolinic acid injections into the left motor area [193]. PET imaging detected a significant increase in the ^18^F-FENM uptake 24 h and 72 h after excitotoxic lesions compared to the control group. So, although FENM showed only a moderate affinity for the PCP site, the tracer appeared suitable to track NMDAR activation in neurodegenerative and neurological diseases.

### 4.2. FENM as an Anxiolytic Agent in PTSD

Besides their potentialities in neurodegenerative diseases, NMDAR antagonists have shown efficacy in a variety of neuropsychiatric disorders. Ketamine, notably, is widely known for its rapid-acting long-lasting antidepressant effect and its efficacy in treatment-resistant depression [106,194,195]. It also prevented stress-induced behavioral despair and attenuated learned fear when administered prior to stress [196,197,198], suggesting a potential efficacy in rodent models of PTSD. FENM has been investigated in this indication. FENM and memantine were administered in rats submitted to a battery of behavioral assays, including prepulse inhibition, open-field locomotion, a light–dark test, forced swimming, and cued fear conditioning [199]. When administered at different timepoints prior to cued fear conditioning or extinction training, FENM reduced fear behavior in a long-lasting and dose-specific manner. It also attenuated learned fear when administered acutely, prior to the conditioning, indicating that it may also be used as a resilience-enhancing prophylactic. Importantly, FENM did not alter sensorimotor gating during prepulse inhibition or locomotion in the open field, contrarily to memantine that reduced the startle response and locomotion 30 min after injection. These data suggested that FENM could be devoid of non-specific side effects [199]. In a PTSD model, the drug was tested as a prophylactic or antidepressant against stress-induced maladaptive behavior, contextual fear conditioning in both male and female mice, and in comparison with ketamine [200]. Given after stress, FENM decreased behavioral despair and reduced perseverative behavior. When administered after re-exposure, FENM facilitated extinction learning. As a prophylactic, FENM attenuated learned fear and decreased stress-induced behavioral despair [200]. Interestingly, ketamine but not FENM increased the expression of c-fos in the CA3 ventral hippocampal area, while both ketamine and FENM attenuated large-amplitude AMPA receptor-mediated bursts in the same area suggesting at least some common neurobiological mechanism [200]. These data outlined the potentialities of FENM as an NMDAR antagonist-acting anxiolytic drug and seeded future development in PTSD.

### 4.3. FENM as a Neuroprotective Agent in AD

FENM was examined, in comparison with memantine, as a neuroprotectant in AD models. Two models were used in parallel, a pharmacological model induced in mice by the intracerebroventricular injection of pre-aggregated/oligomeric amyloid-β_25–35_ (Aβ_25–35_) peptide [201,202] and the APP_swe_/PSEN1^∂E9^ transgenic line [203]. Both memantine and FENM showed symptomatic anti-amnesic effects at a low mg/kg dose range in Aβ_25–35_-treated mice submitted, one week after the peptide injection, to a battery of behavioral tests: spontaneous alternation, passive avoidance, object recognition, place learning in the water-maze, and the Hamlet test measuring topographic memory. Interestingly, FENM was not amnesic when tested alone at a high dose, 10 mg/kg, contrary to memantine. When drugs were injected in the same low mg/kg dose range but once-a-day during one week, they prevented Aβ_25–35_-induced memory deficits, oxidative stress (lipid peroxidation, cytochrome c release), inflammation (IL-6, TNFα increases; GFAP and Iba1 immunoreactivity in the hippocampus and cortex), and apoptosis and cell loss (Bax/Bcl-2 ratio; cell loss in the hippocampus CA1 area) [202]. FENM effects, notably on neuroinflammation, were more robust than observed with memantine. A second study examined FENM efficacy when the drug was subcutaneously infused using an osmotic minipump one week after the Aβ_25–35_ injection in mice. Deficits in spontaneous alternation and object recognition were prevented by infused FENM [204]. Similar effects were observed with daily intraperitoneal FENM or memantine treatments. Animals infused at 0.1 mg/kg/day showed the prevention of Aβ_25–35_-induced neuroinflammation, oxidative stress, and apoptosis. Aβ_25–35_ provoked, in hippocampal homogenates or synaptosomes, a decrease in the PSD-95 level, an increase in the GluN2A subunit with a decreased phosphorylation level, and no change in GluN2B but an increase in its phosphorylation. The FENM infusion attenuated Aβ_25–35_-induced alteration in PSD-95, GluN2A, and P-GluN2B levels but not P-GluN2A, showing a direct regulation of NMDAR in AD mice. GluN2D levels were unchanged whatever the treatment. In 10-month-old APP_swe_/PSEN1^∂E9^ and wildtype control mice, FENM was administered for four weeks, either by daily intraperitoneal injections at 0.3 mg/kg or chronic subcutaneous infusion at 0.1 mg/kg/day using an osmotic minipump. Animals were then examined for spatial working memory, neuroinflammation, apoptosis, amyloid load markers, and synaptic LTP in hippocampal slices [204]. Both infused and repeated intraperitoneal FENM treatments attenuated the behavioral deficits, microglial activation, resulting increases in cytokines TNFα and IL-6, and increase in Bax levels in the mouse hippocampus. Both the soluble and insoluble Aβ_1–40_ and soluble Aβ_1–42_ cortical levels were attenuated by the treatments. The alteration of long-term potentiation maintenance in CA1 of the hippocampus of APP_swe_/PSEN1^∂E9^ mice was also improved by the treatments. These data confirmed that a post-symptomatic treatment with FENM allowed for the significant prevention of AD pathology in a transgenic mouse model.

## 5. Conclusions

We provided an overview of the research progress on the neurobiological effects of NMDARs, in terms of composition, function, and modulation, which are associated with many neurological and psychiatric disorders. Among them, neurodegenerative pathologies, namely Alzheimer’s, Huntington’s, and Parkinson’s disease, occur due to or are amplified by alterations of NMDAR activity responsible for a chronic excitotoxic status, and lead to specific alterations, as observed in AD with Aβ directly affecting the excitatory synapse. NMDAR antagonists, and mainly uncompetitive antagonists targeting the PCP site, have led to major clinical breakthroughs, including memantine in AD, ketamine in anesthesia or major depression, and amantadine in PD. Novel drugs are still in development and we particularly focused on FENM, a memantine derivative that presents superior neuroprotective activity in AD as compared to memantine and superior anxiolytic properties in PTSD as compared to ketamine. In AD preclinical models, the drug has been shown to alleviate amyloid toxicity acutely in a pharmacological model and chronically when administered after the symptoms’ onset in a transgenic model. Moreover, several administration pathways were used, suggesting different possible galenic forms in clinics. Longitudinal studies and precise analyses of the drug impact on the brain’s morphology and plasticity, particularly in Aβ and tau models of AD, are still in progress, but FENM is expected to enter clinical trials for the second semester of 2024.

## Figures and Tables

**Figure 1 ijms-25-03733-f001:**
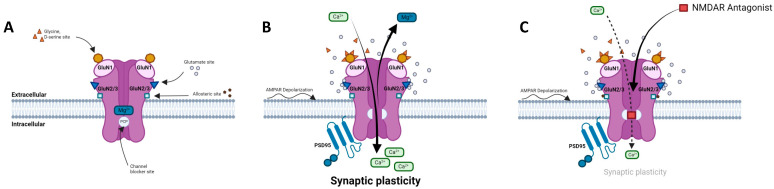
*N*-methyl-d-aspartate receptor (NMDAR) and its ligand binding sites. (**A**) NMDAR at resting potential is blocked by Mg^2+^. (**B**) NMDAR is activated by AMPAR-induced depolarization and binding of both Glu and Gly/D-Ser. Activation results in opening of channel and allows for voltage-dependent release of Mg^2+^ out of ionophore and Ca^2+^ influx into cell, inducing activation of PSD-95, signaling kinases, and mediating synaptic plasticity. (**C**) NMDAR uncompetitive antagonist binds PCP site and partially blocks influx of Ca^2+^ thus preventing neuronal membrane depolarization and downstream signaling mechanisms. Adapted from [15,26,27,33].

**Figure 3 ijms-25-03733-f003:**
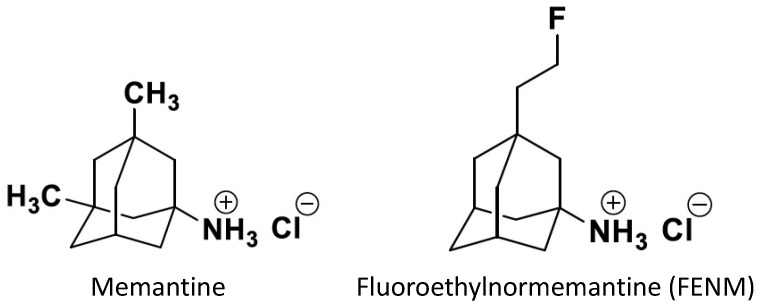
Structures of memantine and fluoroethylnormemantine. The PET radiotracer carries a ^18^F positron-emitting isotope [191,192].

## Data Availability

Not applicable.

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
