# Peer review of "Targeting N-Methyl-d-Aspartate Receptors in Neurodegenerative Diseases"

_ijms, 2024, doi:10.3390/ijms25073733_

Round 1

Reviewer 1 Report

Comments and Suggestions for Authors

Manuscript entitled Targeting N-methyl D-asparate receptors in ND is recommended for minor revision. However, following comments must be addressed before submitting revised manuscript.

Abstract and conclusion part must be rewritten for better understanding of review concept.

Include recent references.

Draw structure of all molecules which are in clinical trials.

Figures must be cross varify with recent references.

Include future perspective of the study.

It is recommended not to include unpublished data in the review.

1. What is the rationality of this review compare to other published papers?

2. Include clinical aspects based on recent research on the topic

3. It is equally important for the readers to know chemical structures. Draw structure of all molecules which are in clinical trials.

4. What specific improvements should the authors consider regarding the

methodology? What further controls should be considered?

5. Abstract and conclusion part must be rewritten for better understanding of review concept.

6. Cross check the references with authenticated sources.

7. Figures must be updated using recent literature.

8. Some of the sentences are repeated. Please check for plagiarism percentage.

9. Avoid including unpublished data (page 11 and 12).

For example: GluN2D levels were unchanged whatever the treatment. In 10-month-old APPswe/PSEN1∂E9 and wildtype control mice, FENM was administered forfour weeks, either by daily intraperitoneal injections at 0.3 mg/kg or chronic subcutaneous infusion at 0.1 mg/kg/day using osmotic minipump. Animals were then examined for spatial working memory, neuroinflammation, apoptosis, amyloid load markers, and synaptic LTP in hippocampal slices [Carles et al., submitted].

Comments on the Quality of English Language

Moderate editing is required.

Author Response

Manuscript entitled Targeting N-methyl D-asparate receptors in ND is recommended for minor revision. However, following comments must be addressed before submitting revised manuscript.

The thank the reviewer for his positive evaluation and therafter explain how we addressed his critics.

Abstract and conclusion part must be rewritten for better understanding of review concept.

Include recent references.

Both abstract and conclusions were corrected to precisely describe the review content. 29 recent references were added to improve the literature cited.

Draw structure of all molecules which are in clinical trials.

The structures have been added in the Table.

Figures must be cross varify with recent references.

The figures were referenced as requested.

Include future perspective of the study.

Perspectives have been added in the conclusion of the review.

It is recommended not to include unpublished data in the review.

Unpublished data were withdrawn in paragraph 3.3.

  1. What is the rationality of this review compare to other published papers?

This review provides a concise assessment of the role and impacts of the NMDA system in neurodegenerative diseases, particularly in terms of structure, distribution, localization and regulation of the glutamatergic synapse in neurodegenerative diseases. It also highlights a new uncompetitive antagonist whose very recent results show the interest in these indications. This is now clearly stated in the abstract and introduction.

  1. Include clinical aspects based on recent research on the topic.

Clinical aspects are considered in the review, notably in paragraphs 3.1 to 3.3.

  1. It is equally important for the readers to know chemical structures. Draw structure of all molecules which are in clinical trials.

The chemical structures of all cited molecules are not shown in the table and text.

  1. What specific improvements should the authors consider regarding the methodology? What further controls should be considered?

This comment is out of the scope of a review but more adequate in original research articles. We did not see on what part of our text this comment applies.

  1. Abstract and conclusion part must be rewritten for better understanding of review concept.

Both abstract and conclusions were corrected to precisely describe the review content.

  1. Cross check the references with authenticated sources.

All references were carefully checked.

  1. Figures must be updated using recent literature.

The figures were updated and referenced as requested.

  1. Some of the sentences are repeated. Please check for plagiarism percentage.

The text was carefully checked for, proper english and for repetitions and checked for plagiarism with the university software Compilatio Magister. The percentage of similarity is now 6%.

  1. Avoid including unpublished data (page 11 and 12).

For example: GluN2D levels were unchanged whatever the treatment. In 10-month-old APPswe/PSEN1∂E9 and wildtype control mice, FENM was administered forfour weeks, either by daily intraperitoneal injections at 0.3 mg/kg or chronic subcutaneous infusion at 0.1 mg/kg/day using osmotic minipump. Animals were then examined for spatial working memory, neuroinflammation, apoptosis, amyloid load markers, and synaptic LTP in hippocampal slices [Carles et al., submitted].

These data were presented in the Society for Neuroscience meeting and the online abstract is referenced, as it is accepted by the neuroscientific community. Unpublished and undisclosed data were deleted.

Reviewer 2 Report

Comments and Suggestions for Authors

In this review, Carles and colleagues discuss the importance of targeting N-methyl-D-aspartate receptors in neurodegenerative diseases. The manuscript is well-written and interesting. However, some points should be addressed.

-       Title: The Authors discuss data from preclinical models of PTSD. However, PTSD is not recognized as a neurodegenerative disorder.

-       A discussion about the impact of sex in this context is necessary. Indeed, both PTSD and AD are more prevalent in women rather than in men. Moreover, sex-differences in NMDA receptors functioning are reported both in physiological and pathophysiological conditions (PMID: 32173404; PMID: 37293561). Thus, it is imperative to test the efficacy of NMDA antagonists in rodents of both sexes. The Authors must discuss this pivotal point.

-       2.4 NMDAr modulators: D-AP5 is an antagonist an not an agonist. Please correct Among competitive NMDAR agonists, the highly selective proto-type drug is D-AP5.ù

-       Page 11:FENM or memantine were administered in rats submitted to a battery of behavioral assays, including paired-pulse inhibition.” I think the Authors meant pre- pulse inhibition.

-       A moderate editing of English is suggested.

Comments on the Quality of English Language

Moderate editing

Author Response

Reviewer 2

Comments and Suggestions for Authors

In this review, Carles and colleagues discuss the importance of targeting N-methyl-D-aspartate receptors in neurodegenerative diseases. The manuscript is well-written and interesting. However, some points should be addressed.

We thank the reviewer for his positive evaluation. We thereafter answered his critics and comments.

-       Title: The Authors discuss data from preclinical models of PTSD. However, PTSD is not recognized as a neurodegenerative disorder.

The purpose of the review is on the role of NMDA receptors in neurodegenerative disorders, not psychiatric diseases. This was more clearly stated in the introduction. However, FENM, the novel investigational drug that we present in the last part of the review as a prototypic  example of NMDAR modulator with therapeutic potentials, has shown preclinical effects in PTSD as well as in AD. It would be incomplete to present only the FENM effects in AD. We however shortened the paragraph 4.2, describing FENM's effects in PTSD (and as a PET radiotracer as well) to maintain the focus on NMDA receptors in neurodegenerative diseases as stated by the title. The review now focus on neurodegenerative disorders in part 1 and part 2 and the paragraph on PTSD in part 3 was considerably shortened.

-       A discussion about the impact of sex in this context is necessary. Indeed, both PTSD and AD are more prevalent in women rather than in men. Moreover, sex-differences in NMDA receptors functioning are reported both in physiological and pathophysiological conditions (PMID: 32173404; PMID: 37293561). Thus, it is imperative to test the efficacy of NMDA antagonists in rodents of both sexes. The Authors must discuss this pivotal point.

The reviewer is right in pointing out the importance of gender differences. As this is particularly crucial in neurodegenerative diseases like AD, we added a paragraph in the 3.1 part, notably by discussing a pertinent clinical study identifying differences in L-Serine contents, so indirectly in NMDAR activation, in the brain of male and female patients.

-       2.4 NMDAr modulators: D-AP5 is an antagonist an not an agonist. Please correct Among competitive NMDAR agonists, the highly selective proto-type drug is D-AP5.

Sorry for the typo. Corrected.

-       Page 11: “FENM or memantine were administered in rats submitted to a battery of behavioral assays, including paired-pulse inhibition.” I think the Authors meant pre- pulse inhibition.

Sorry for the mistake. Corrected.

Round 2

Reviewer 2 Report

Comments and Suggestions for Authors

The Authors have addressed the points I raised. 

Comments on the Quality of English Language

minor editing